# The Stability of U(VI) and As(V) under the Influence of pH and Inorganic Ligands

Qingshan Tian [1], Peng Wang [2], Yunxiang Huang [1,*], Bo Zhang [3,*] and Wentao Jiao [3]

1 College of Resources and Environment, Hunan Agricultural University, Changsha 410128, China
2 China Haohua North Chem, No. 19 Xiaoying Road, Chaoyang District, Beijing 100101, China
3 Research Center for Eco-Environmental Sciences, Chinese Academy of Sciences, No. 18 Shuangqing Road, Haidian District, Beijing 100085, China
* Correspondence: yxhuang63@163.com (Y.H.); zhangbo@rcees.ac.cn (B.Z.)

**Abstract:** Uranium and arsenic are two pollutants commonly found in groundwater near uranium mines. However, the reactivity of U(VI) and As(V) need to be carefully investigated to better understand their fate and transport in the environment. In this study, the reactivity of U(VI) and As(V) were studied under various pH, bicarbonate, and $Ca^{2+}$ levels. In air-equilibrated systems, the reactivity of U(VI) and As(V) generally decreased with the increase in pH, as evidenced by the solubility of U(VI) and As(V) increasing along with the elevation of pH. At pH = 8, 44.70% and 37.81% of initially added U(VI) and As(V) remained soluble. The addition of 1 mM of bicarbonate increased the reactivity of U(VI) and As(V) at mild acidic to neutral pH; however, the presence of bicarbonate significantly increased the solubility of U(VI) at mild alkaline condition, as nearly all U(VI) remained soluble at pH values of 8 and 9. After the addition of $Ca^{2+}$, the solubility of both U(VI) and As(V) decreased at mild acidic to neutral pH ranges; however, the addition of $Ca^{2+}$ markedly increased the soluble percentages of U(VI) at neutral pH, in which the condition 97.81 ± 2.95% of U(VI) remained soluble. Comparatively, only 36.13 ± 4.98% and 1.69 ± 1.08% of U(VI) were soluble at the same pH in air-equilibrated and bicarbonate systems. Our study demonstrated that U(VI) and As(V) are less reactive at neutral to alkaline conditions. Furthermore, the addition of bicarbonate and $Ca^{2+}$ can further reduce the reactivity of U(VI) and As(V) at neutral to alkaline conditions. The findings of this study contribute to a deeper understanding of the fate and transport of U(VI) and As(V) in groundwater and could aid in better designing of U(VI) and As(V) removal processes.

**Keywords:** groundwater; heavy metal; precipitation; phreeqc; water chemistry

## 1. Introduction

Uranium and arsenic are two commonly encountered pollutants in groundwater systems. Previous studies have demonstrated that U and As could enter groundwater through both natural and anthropogenic processes. For instance, groundwater with elevated concentrations of U and As was found in Hetao Basin, China, as a result of the weathering of U- and As-containing bedrock in the local mountains [1]. Additionally, it was found that U and As concentrations in groundwater near abandoned mine wastes located in Northwestern Arizona, U.S., exceeded the maximum contaminant levels (MCLs) set by the U.S. EPA [2]. Both U and As are highly toxic and can cause severe health issues, such as skin lesion, liver cancer, and kidney failure [3,4]. Considering that groundwater is increasingly being used as a source of drinking water in these rural areas, it is of paramount importance to understand and be able to predict the environmental behavior of U and As in groundwater.

The mobilities of U and As in groundwater are closely correlated with their redox state and groundwater chemical compositions. Uranium typically exists as U(IV) or U(VI) in the environment. While U(IV) tends to forms sparingly soluble precipitates in anoxic environments and can subsequentially be immobilized [5,6], U(VI) generally exists as

soluble uranyl ($UO_2^{2+}$) or its associated complexes in oxic environments. In addition, it is well known that U(VI) could combine with carbonate [7], which is an anion ubiquitously present in groundwater, to form uranyl–carbonato complexes. Compared to uranyl ion, the uranyl–carbonato complex is more stable and has lower reactivity [8]. Moreover, an even more stable and soluble Ca–uranyl–carbonato ternary complex could form with $Ca^{2+}$, which is also a common groundwater cation [9–11]. The formation of binary and ternary complexes could significantly enhance the solubility of U(VI) compared to non-complexed forms and increase the mobility of uranium in groundwater. The fate and transfer of As is also closely controlled by its redox state [12,13]. In an anoxic environment, As exists as $H_3As(III)O_3$, which is a neutral molecule and has higher mobilities. In an oxic environment, As predominantly exists as arsenate anions ($HAs(V)O_4^{2-}$), which could be adsorbed by iron-(oxy)hydroxides [14], clay minerals, and natural organic matter [15]. Generally speaking, in oxic groundwater, U(VI) and As(V) are the predominant species for uranium and arsenic, respectively [16].

Other than carbonate, U(VI) could complex with phosphate and produce a non-soluble precipitate in a wide pH range. It has been demonstrated that $UO_2^{2+}$ could complex with $PO_4^{3-}$ to form chernikovite ($H_3O(UO_2)(PO_4)\cdot3H_2O$), a type of precipitate [17,18] which can greatly limit the solubility of U(VI) in groundwater. Furthermore, the stability of the $UO_2^{2+}$-$PO_4^{3-}$ system is also controlled by pH, and the concentration of dissolved inorganic carbon, that is, the solubility of $UO_2^{2+}$, is higher in mild alkaline to neutral conditions, and the presence of dissolved $CO_3^{2-}$ could further increase the solubility of $UO_2^{2+}$ [10]. It has also been demonstrated that the reaction pathways, i.e., the sequence at which the $UO_2^{2+}$ and $PO_4^{3-}$ are added, also played an important role in the product formed. Specifically, at mild acidic conditions (4 < pH < 6), the Ca-$UO_2^{2+}$-$PO_4^{3-}$ may dominate the product if the $UO_2^{2+}$ and $PO_4^{3-}$ were added simultaneously; however, if $PO_4^{3-}$ was added prior to the addition of $UO_2^{2+}$, that could lead to the precipitation or incorporation of $UO_2^{2+}$ to the Ca-$PO_4$ solids [11]. A recent study demonstrated that $UO_2^{2+}$ could form insoluble solids with $PO_4^{3-}$ at pH values of 3 and 7, even in the presence of high concentrations of $Ca^{2+}$ and $CO_3^{2-}$ [10]. Arsenate ($AsO_4^{3-}$) and phosphate have similar chemical structures and are analogs of each other [19]; therefore, U(VI) can form both soluble and insoluble complexes with As(VI) [20,21]. Considering that phosphate can react with U(VI) to form insoluble precipitates, it is intriguing to examine the reactivity of U(VI) and As(V) to form precipitates. Such reactivity, i.e., the tendency to form insoluble precipitates, is of vital importance to determine the ability of U(VI) and As(V) to transfer along with the flow of groundwater.

The objective of this study is to determine the reactivity of U(VI) and As(V) from mild acidic to mild alkaline conditions under the presence of a common inorganic ligand of U(VI). Specifically, we want to investigate the stability of U(VI) and As(V) in slightly alkaline conditions (7 < pH < 10), which has been reported in U(VI)-As(V) co-existing groundwater previously. The wet chemistry experiments were conducted with varying pH and concentrations of $CO_3^{2-}$ and $Ca^{2+}$ in the first place, then the measured soluble U(VI) and As(V) concentrations were used to calibrate model simulations. The findings of our study could contribute to better understanding the chemical behavior of U and As in groundwater and ultimately led to better pollution control strategies.

## 2. Materials and Methods

### 2.1. Chemicals

The chemicals used in this study were of ACS grade or better. A 10 mM uranyl nitrate $UO_2(NO_3)_2$ (Sinopharm, Beijing, China) stock solution was prepared in ultrapure water (18.2 MΩ·cm resistivity). A 20 mM arsenate stock solution was prepared in ultrapure water using sodium arsenate (Sigma-Aldrich, St. Louis, MO, USA). Dilute sodium hydroxide (NaOH) solution and/or nitric acid solutions were used to adjust the pH of the solutions to the target values. In order to study the effect of bicarbonate and $Ca^{2+}$ on the reactivity

of uranyl and arsenate, $NaHCO_3$ and $Ca(NO_3)_2$ (Sinopharm, China) were added as the sources of bicarbonate and $Ca^{2+}$, respectively.

### 2.2. Wet Chemistry Experiments

The wet chemistry experiments were carried out at room temperature to study the reactivity of U(VI) and As(V) at different pH values and at different concentration levels of bicarbonate and $Ca^{2+}$. In this study, the reactivity is explicitly used to refer to the tendency of U(VI) and As(V) to form precipitates.

A pH range of 4.0–9.0 is selected because it has been reported that groundwater near uranium mines and disposal sites have similar pH values [22,23]. All experiments were carried out in 40 mL brown VOC vials. According to the specific conditions of each group of experiments, different amounts of U(VI), As(V), $NaHCO_3$, and $CaCl_2$ were added to a $KNO_3$ solution to initiate the reaction (see Table 1 for experimental composition). Three replicates were set for each group of experiments, and control experiments without uranium and arsenate were carried out at the same time to evaluate the removal of uranium or arsenate in the absence of other substances.

**Table 1.** List of experimental conditions (the concentrations were given in mM).

| No. | Combination | U(VI) | As(V) | $NaHCO_3$ | $Ca^{2+}$ | $KNO_3$ |
|---|---|---|---|---|---|---|
| 1 | U(VI) + $KNO_3$ | 0.05 | - | - | - | 1 |
| 2 | As + $KNO_3$ | - | 0.1 | - | - | 1 |
| 3 | U(VI) + As + $KNO_3$ | 0.05 | 0.1 | - | - | 1 |
| 4 | U(VI) + As + $NaHCO_3$ + $KNO_3$ | 0.05 | 0.1 | 5 | - | 1 |
| 5 | U(VI) + As + $NaHCO_3$ + $Ca^{2+}$ + $KNO_3$ | 0.05 | 0.1 | 5 | 1 | 1 |

The concentration of bicarbonate and $Ca^{2+}$ could appreciably affect the stability of soluble U(VI) in groundwater, as bicarbonate and U(VI) could form more soluble uranyl–carbonato complexes, and $Ca^{2+}$ and bicarbonate could form even more soluble Ca–uranyl–carbonato complexes with U(VI). Therefore, we tested the reactivity of U(VI) and As(V) in three different conditions, which are: (1) the air-equilibrated condition, i.e., the control group, in which the bicarbonate concentration was allowed to reach equilibration with atmospheric $CO_2$, but no additional bicarbonate or $Ca^{2+}$ were added; (2) the bicarbonate group, in which additional bicarbonate was added to reach the concentration of 5 mM in experimental settings; (3) the Ca-bicarbonate group, in which both $Ca^{2+}$ and bicarbonate were added to reach the concentration of 1 mM and 5 mM, respectively.

Throughout the experiment, no efforts were undertaken to remove the dissolved $CO_2$ from air. In addition, for experimental groups with added bicarbonate, the pH was typically adjusted using dilute NaOH or $HNO_3$ within 2 min, and then the vials were quickly sealed with headspace volume of ca. 1 mL, to minimize the loss of dissolved $CO_2$.

The pH values of the solutions were adjusted using diluted NaOH and $HNO_3$ solutions, respectively. The concentrations of these two solutions used were 1% (m/m) and were added in a dropwise manner. The pH values were monitored using a pH meter (FiveEasy, Mettler Toledo, Columbus, OH, USA).

### 2.3. Measuring the Concentration of U and As

The solution is exposed to air only during initial loading, pH adjusting, and sampling. Samples were collected from the reaction vessel at pre-determined sampling time points. Samples for measurement of dissolved U and As were filtered using 0.22 μm polycarbonate membrane syringe filters. Then, 50 μL of filtrate was taken, with the addition of 250 μL of concentrated $HNO_3$ for acidification, and 4.7 mL of ultrapure water for dilution. The concentrations of U and As in the diluted samples were measured using ICP-MS (Nexion 300, Pekin-Elmer, Waltham, MA, USA).

### 2.4. Simulation of Groundwater Compositions

The speciation of soluble U(VI) and As(V), as well as the solid phases that would form during the experiments, were simulated using PHREEQC (Version 3.3.12.12704). The database from the Lawrence Livermore National Laboratory (LLNL) was used to facilitate the calculations. In addition to the LLNL database, the formation constants of Ca–uranyl–carbonato complexes were adopted from a previous study [10].

## 3. Results and Discussion

### 3.1. The Effect of pH on the Reactivity of U(VI) and As(V)

The effect of pH on the solubility of U(VI) and As(V) was firstly investigated in air-equilibrated systems. Assuming the partial pressure of $CO_2$ ($P_{CO2}$) is 380 ppm, then it can be calculated that the $TOTCO_3$ concentration is near 0.1 mM. From Figure 1, it can be seen that As(V) remained soluble when there was no U(VI) present throughout the entire studied pH range, as evidenced by the near 100% of added As(V) that remained soluble from pH = 4 to pH = 9. U(VI) remained soluble at mild acid and alkaline conditions, as suggested by the fact that nearly 100% of U(VI) remained soluble. However, the solubility of U(VI) was low at near-neutral conditions (Figure 2). Specifically, only 21.46 ± 0.09% and 54.91 ± 0.06% of U(VI) remained in solution at pH of 5 and 6, respectively. It is likely that $UO_2^{2+}$ hydrolyzed to form insoluble solids such as $UO_2(OH)_2$, which has the net effect of reducing the solubility of U(VI) at neutral pH conditions.

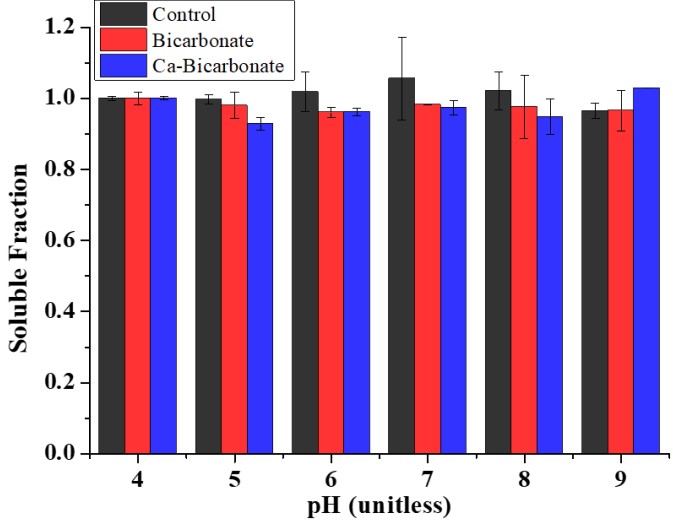

**Figure 1.** The soluble percentages of As(V) with no addition of U(VI) at pH 4~9. Control: the air-equilibrated group, bicarbonate: the group with 5 mM addition of bicarbonate, Ca-Bicarbonate: the group with 5 mM bicarbonate and 1 mM $Ca^{2+}$ addition.

When U(VI) and As(V) were added simultaneously, it can be seen that the soluble percentages of both U(VI) and As(V) decreased compared to the control groups (Figures 3 and 4). Specifically, the soluble percentages of As(V) varied from 30.58 ± 4.05% at pH = 4 to 37.81 ± 3.91% at pH = 8 (Figure 3). Similarly, the soluble percentages of U(VI) varied from 11.10 ± 0.25% at pH = 4 to 44.70 ± 10.91% at pH = 8 (Figure 4). For both U(VI) and As(V), their solubility were closely related to the pH of the solution. If we plot the soluble percentages of the U(VI) and As(V) against the pH of the solution, then it can be found that the linear regression coefficients are 0.78 for U(VI) and 0.83 for As(V), respectively (see Figure 5, more fitting results are given in Figure S1 and Table S1 in the Supplementary Material). These results clearly indicated that in the air-equilibrated system, the reactivity of U(VI) and As(V) are related to the pH of the solution. At lower pH, the U(VI) tends to react with As(V) to form solids; however, at higher pH region, especially

at mild alkaline conditions, the reactivity of U(VI) and As(V) lowered and led to higher percentages of soluble U(VI).

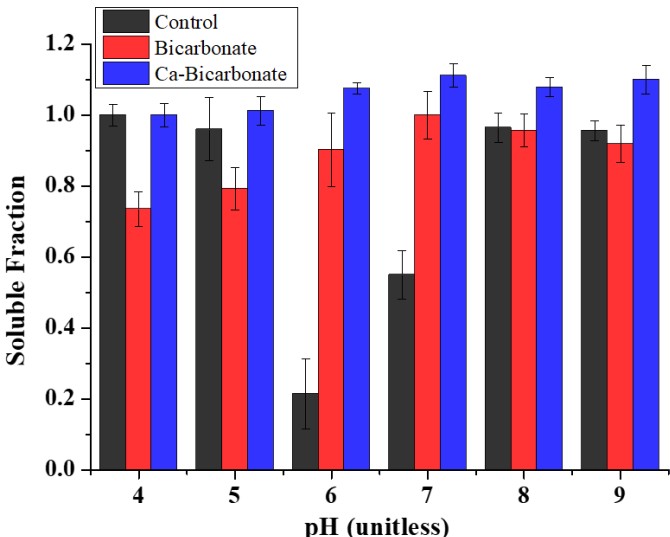

**Figure 2.** The soluble percentages of U(VI) with no addition of As(V) at pH 4~9. Control: the air-equilibrated group, bicarbonate: the group with 5 mM addition of bicarbonate, Ca-Bicarbonate: the group with 5 mM bicarbonate and 1 mM $Ca^{2+}$ addition.

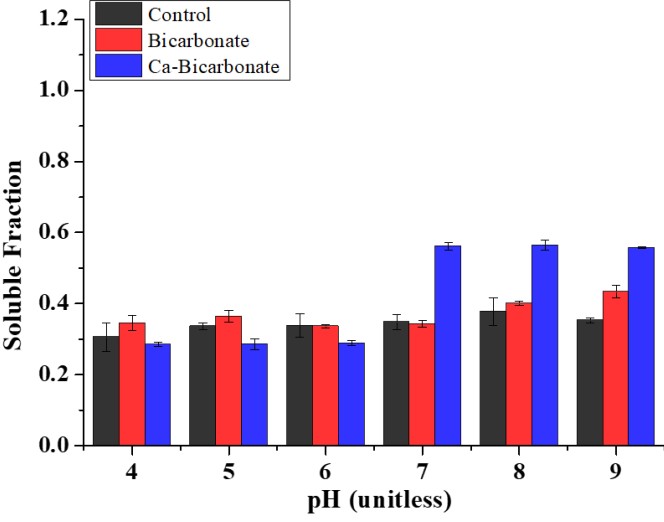

**Figure 3.** The soluble percentages of As(V) with the addition of U(VI) at pH 4~9. Control: the air-equilibrated group, bicarbonate: the group with 5 mM addition of bicarbonate, Ca-Bicarbonate: the group with 5 mM bicarbonate and 1 mM $Ca^{2+}$ addition.

### 3.2. Effect of Bicarbonate on Reactivity of U(VI) and As(V)

In the second set of experiments, $NaHCO_3$ was added to investigate the effect of bicarbonate on the reactivity of U(VI) and As(V). The presence of $CO_3^{2-}$ did not alter the solubility of $AsO_4^{3-}$ compared to the air-equilibrated system, which is evidenced by the fact that nearly all As(V) remained soluble from pH = 4 to pH = 9 (see Figure 1). Contrary to As(V), the addition of bicarbonate apparently increased the solubility of U(VI). Unlike in the air-equilibrated system, essentially all the added U(VI) remained soluble after 1 mM of bicarbonate was added, even at pH of 6 and 7, at which point noticeable U(VI) hydrolysis occurred in the air-equilibrated system (see Figure 2). For U-As binary systems, the presence of bicarbonate slightly increased the soluble As(V) percentages at mild acidic and mild alkaline conditions. It can be seen in Figure 3 that the soluble As(V) percentages increased

from $30.58 \pm 4.05\%$ to $34.50 \pm 2.06\%$ (pH = 4.0), $33.63 \pm 0.97\%$ to $36.43 \pm 1.65\%$ (pH = 5.0), $37.81 \pm 3.90\%$ to $40.08 \pm 0.67\%$ (pH = 8.0), and $35.27 \pm 0.59\%$ to $43.30 \pm 1.78\%$ (pH = 9.0). In mild alkaline conditions, the addition of bicarbonate also led to higher soluble U(VI) percentages compared to the air-equilibrated control group; according to Figure 4, the soluble U(VI) percentage increased from $44.70 \pm 1.69\%$ to $95.32 \pm 1.67\%$ (pH = 8.0) and $36.16 \pm 6.03\%$ to $97.33 \pm 1.35\%$ (pH = 9.0).

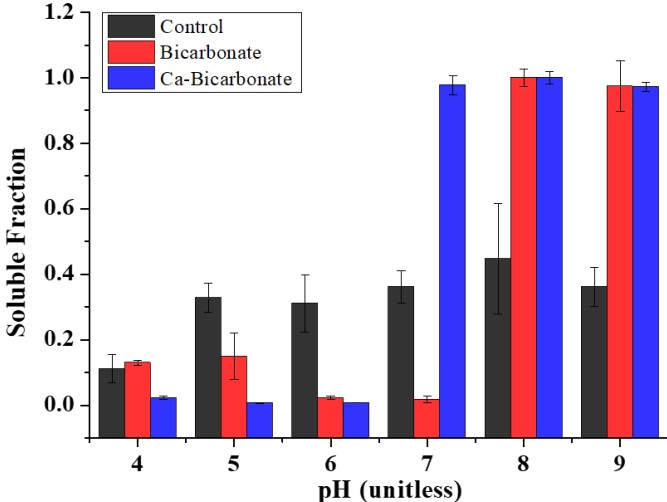

**Figure 4.** The soluble percentage of U(VI) with the addition of As(V) at pH 4~9. Control: the air-equilibrated group, bicarbonate: the group with 5 mM addition of bicarbonate, Ca-Bicarbonate: the group with 5 mM bicarbonate and 1 mM $Ca^{2+}$ addition.

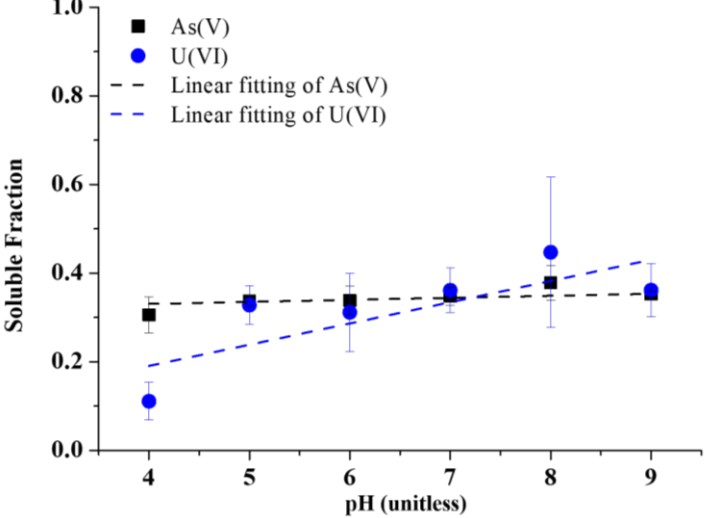

**Figure 5.** The relationship between pH and the soluble percentages of U(VI) and As(V) in air-equilibrated binary system.

### 3.3. Effect of $Ca^{2+}$ and $CO_3^{2-}$ on Reactivity of U(VI) and As(V)

To investigate the effect of $Ca^{2+}$ and $CO_3^{2-}$ on the reactivity of U(VI) and As(V), these two ions were simultaneously added to the solution that contains U(VI), As(V), or both U(VI) and As(V), in the form of $CaCl_2$ and $NaHCO_3$.

From Figure 1, it can be seen that similar to the air-equilibrated group and the bicarbonate-addition group, the soluble As(V) percentages did not change noticeably in the $Ca^{2+}$-$CO_3^{2-}$ group. On the other hand, the soluble U(VI) concentrations were all increased compared to air-equilibrated group, which could be interpreted by the formation of a stable and soluble $Ca$-$UO_2^{2+}$-$CO_3^{2-}$ complex.

In U(VI)-As(V) binary systems, the solubility of As(V) decreased at mild acidic pH range (see Figure 3) compared to the air-equilibrated group and the bicarbonate group. Considering the similarities between the chemical structures of arsenate and phosphate ions, we speculated that the decrease in As(V), after the addition of $Ca^{2+}$, is due to the formation of a $Ca-AsO_4$ precipitate. Similarly, the soluble percentage of U(VI) was also lowered in U-As binary systems after the addition of $Ca^{2+}$ (see Figure 4). The soluble percentages of U(VI) were 2.20 ± 0.49%, 0.62 ± 0.18%, and 0.73 ± 0.04%, for pH values of 4, 5 and 6, respectively.

Both soluble U(VI) and As(V) percentages increased at neutral to alkaline conditions after the addition of $Ca^{2+}$ and $CO_3^{2-}$. The soluble percentages of As(V) were 56.20 ± 1.07%, 56.43 ± 1.41%, and 55.70 ± 0.23% at pH values of 7, 8, and 9, and the soluble U(VI) accounted for 97.81 ± 0.95%, 98.87 ± 1.97%, and 97.29 ± 1.43% of total U(VI) at the same pH range. It is hypothesized that at higher pH condition, the formation of the Ca–uranyl–carbonato complex is more favorable, which can markedly increase the solubility of U(VI). Meanwhile, the formation of the Ca–uranyl–carbonato complex also has the net effect of decreasing the concentration of $Ca^{2+}$ and thereby decreasing the propensity of $Ca-AsO_4$ precipitate formation.

It is worth noting that when $Ca^{2+}$ and bicarbonate were simultaneously present, the soluble U(VI) increased appreciably at pH = 7, which was not the case in the air-equilibrated and the bicarbonate systems. This fact likely was also a result of the formation of the Ca–uranyl–carbonato complex. Considering that the formation of this ternary complex system led to higher soluble U(VI) percentages at neutral pH, it can be inferred that the presence of $Ca^{2+}$ can lead to higher U(VI) solubilities at neutral pH and greatly facilitate the transfer of U(VI) and As(V) in groundwater.

### 3.4. Comparison of U(VI)'s Solubility in Uranium–Arsenate and Uranium–Phosphate Systems

A previous study investigated the solubility of U(VI) in a uranyl–phosphate (U-P) system at similar concentration ranges, which enabled us to compare the reactivity of arsenate and phosphate towards uranyl at environmentally relevant concentrations and pH [11]. For U-P systems, 6.30%, 31.6%, and 79.4% of initially added U(VI) remained soluble at pH values of 4, 6, and 8 with no bicarbonate addition conditions. When 1 mM of bicarbonate was provided, the soluble percentages of U(VI) at pH = 4 slightly increased to 7.08%; however, essentially all U(VI) remained soluble in higher pH conditions of 6 and 8. However, for the U(VI)-As(V) system, the soluble percentages of U(VI) were 12.96 ± 0.73%, 21.90 ± 0.53%, and essentially 100% at similar pH conditions.

It can be clearly seen that at near-neutral conditions, the solubility of U(VI) in the U-P system is higher than that in As(V) systems. The detailed explanation may require exquisite analysis of solids formed and accurate thermodynamic data with respect to the solid's formation energies, both of which are beyond the scope of this study. However, one possible explanation is the nucleation theory, which is that the U and As tend to form a higher number of small nuclei at pH = 6 than does the U-P system, which allows the formation of higher concentrations of solid precipitates and leads to lower soluble U(VI) at this pH. With the addition of $Ca^{2+}$, essentially all U(VI) precipitated at pH values of 4.5 and 6.8, respectively, and only about 1% of U(VI) remained soluble at pH = 7.5. For the U-As system, only 0.62 ± 0.02% of U(VI) remained soluble at pH = 5; however, the soluble percentages of U(VI) were 97.81 ± 2.90%, 99.83% ± 1.97%, and 97.29 ± 1.48% at pH values of 7, 8, and 9.

These results clearly indicated that U(VI) was less reactive to As(V) than to phosphate. In the Ca-U-P system, $Ca^{2+}$ and phosphate can form hydroxylapatite ($Ca_5(OH)(PO_4)_3$), whose $logK_{sp}$ is −3.22. On the other hand, in the Ca-U-As(V) system, $Ca^{2+}$ and arsenate can form hydroxylapatite ($Ca_3(AsO_4)_2$), whose $logK_{sp}$ value is 17.80 (the detailed speciations of U and As are given in Supplementary Material, see Figure S2, as well as Tables S2 and S3). Comparing these two $logK_{sp}$ values, it is obvious that the solubility of $Ca_5(OH)(PO_4)_3$ is much lower than that of $Ca_3(AsO_4)_2$, which means that more Ca-P precipitate would

form. Previous studies demonstrated that phosphate-containing precipitates, such as hydroxylapatite, are excellent adsorbents for U(VI; therefore, the formed $Ca_5(OH)(PO_4)_3$ precipitate adsorbed a noticeable amount of U(VI) and led to the reduction in soluble U(VI) concentrations. Contrary to the U-P system, the Ca-As precipitate had higher solubility and tended to remain in solution, which would allow the formation of a Ca–uranyl–carbonato complex, which is conducive to maintaining higher soluble U(VI) percentages and lowering the reactivity of U(VI) and As(V).

*3.5. Implications and Limitations of This Study*

The experimental results of this study suggest that U(VI) and As(V) are less likely to form insoluble precipitates in neutral to mild alkaline conditions. The direct implication is that these two species are more likely remain soluble in groundwater and can enter the human body if the groundwater is extracted for potable purposes.

Other than $Ca^{2+}$ and $CO_3^{2-}$, Fe(II) is another ubiquitous ion in the anoxic groundwater environment. In the current study, the effect of Fe(II) on the fate and transport of U(VI) and As(V) was not considered. The findings of our study suggest that U(VI) and As(V) can co-exist in neutral to mild alkaline groundwaters; however, the presence of Fe(II) might directly reduce U(VI) to sparingly soluble U(IV), and the Fe(III)-(oxy)hydroxides formed may serve as a sink to capture As(V). In short, the effect of Fe(II) on the solubilities of U(VI) and As(V) should be further investigated to better understand the environmental behavior of these two pollutants in co-existing conditions.

## 4. Conclusions

In this study, the reactivities of U(VI) and As(V) were investigated under various environmentally relevant conditions. It was found that U(VI) and As(V) were less reactive, i.e., more soluble, at mild alkaline conditions but not at the mild acidic or neutral pH conditions. The presence of bicarbonate decreased the reactivity of U(VI) and As(V) at alkaline conditions; moreover, the further presence of $Ca^{2+}$ could further decrease the reactivity of U(VI) and As(V) in neutral pH, likely as a result of the formation of the Ca–uranyl–carbonato complex. The findings of this study deepened our understanding of the reactivity, fate, and transport of U(VI) and As(V) in groundwater systems and could aid in better designing the U(VI)-As(V) pollution control systems.

**Supplementary Materials:** The following supporting information can be downloaded at: https://www.mdpi.com/article/10.3390/su142012967/s1, Figure S1. The change of soluble U(VI) and As(V) fractions in co-existing conditions as a function of pH. (a) the air-equilibrated group, (b) the bicarbonate group, (c) the Ca-bicarbonate group. Figure S2. The soluble species of As (a) and U (b) in the pH range studied. Table S1. The fitting results of soluble U(VI) and As(V) fractions as function of pH. Table S2. The soluble fraction of As(V) in the studied pH range. Table S3. The soluble fraction of U(VI) in the studied pH range.

**Author Contributions:** Conceptualization, B.Z. and W.J.; methodology, B.Z., P.W. and Q.T.; software, B.Z. and P.W.; validation, Q.T., Y.H. and B.Z.; data curation, Q.T. and P.W.; writing—original draft preparation, B.Z. and Y.H.; writing—review and editing, B.Z. and Y.H.; supervision, W.J.; funding acquisition, B.Z. and W.J. All authors have read and agreed to the published version of the manuscript.

**Funding:** This study is financially supported by the National Natural Science Foundation of China (Nos. 21806176, 41877134).

**Institutional Review Board Statement:** Not applicable.

**Informed Consent Statement:** Not applicable.

**Data Availability Statement:** The data presented in this study are available on request from the corresponding author.

**Conflicts of Interest:** The authors declare no conflict of interest.

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
