# Peer review of "The Stability of U(VI) and As(V) under the Influence of pH and Inorganic Ligands"

_sustainability, doi:10.3390/su142012967_

Round 1
Reviewer 1 Report
This study investigates the interactions between U(VI) and As(V) under a variety of environmental conditions with the aim to establish the fundamental reactivity between the two species. Whilst the environmental motivation for such research is clear, the originality of the study is questionable, and the significant lack of solid state characterisation makes the study feel incomplete. Furthermore, due to limited data, the conclusions drawn here are to fully underpinned by the study and therefore significant further work is required to make this manuscript publishable/novel enough to consider for publication.
Please see my attached PDF file for more detailed comments.

Author Response
We thank the reviewer for the critical and constructive comments. We have prepared a point-to-point response to reviewer's comments for inspection. We hope all the concerns were addressed per to the reviewer's standards.

Reviewer 2 Report
In this paper, the authors investigate the reactivity of uranium and arsenic species in groundwater under various conditions. The paper is well organized and written well. However, the studies are incomplete, and the results are not conclusive.
Comment 1: Abstract: “However, the reactivity of U(VI) and As(V) still needs to be further investigated to predict the fate and transport of these two pollutants.” Please reframe the sentence to “However, the reactivity of U(VI) and As(V) need careful investigation to understand their fate and transport better.”
Comment 2: What was the basis for choosing only the uranium-arsenic pair? Vanadium and iron are prevalent transition metals reported in U contaminated water. Serval other transition metals (Ti, Mn, Co, Cr, Ni, Cu, Zn, etc.) and hydrochemical ions (Na+, K+ Ca2+, Mg2+, NO3-, HCO3-, Cl-, SO42-, etc.) are also present in groundwater. Many background ions can affect uranium dissolution and its fate and transport. Please give more insight. Similarly, why are the studies restricted to calcium and bicarbonates?
Comment 3: What are the reactive species of uranium and arsenic observed in this study? The authors shall tabulate and include them in the revised manuscript.
Comment 4: What was the effect of U and As concentrations on the formation of various uranyl complexes? What is the effect of Eh on solubility?
Comment 5: Are the concentrations of U and As employed in this study relevant to the actual field conditions?
Comment 6: What are the other factors affecting species reactivity under groundwater conditions? Is solubility the only factor that affects reactivity? Several biological and physicochemical factors influence the fate and transport of U. Hydrological, climatic conditions, and mining and leaching technologies can also affect the solubility and fate of transport.
Comment 7: Authors need to provide solid reasons for the reactivity trends observed in different cases. For example, in air-equilibrated samples, the reactivity of the species decreased. The reason given is the formation of soluble species. Is the solubility of species terminating the reactivity?
Comment 8: U concentration in Na-Cl type water is typically higher than Ca-Mg-HCO3- type water. In this case, how did the presence of calcium increase the solubility of uranium in near-neutral pH conditions?
Comment 9: Generally, bicarbonate or carbonate species will form strong metal complexes, leading to a decrease in the reactivity of the corresponding species.
Comment 10: How was the pH of the solutions adjusted in the wet chemistry experiments?
Comment 11: Figures 1 to 4 show that the y-axis is labeled as soluble species percentage, but the values are 0 to 1. Is it the fraction of soluble species?
Comment 12: What is the effect of Fe2+ on the solubility of both species? What could be the expected trend if both Fe2+ and Ca2+ are present in the water in all cases?
Comment 13: The pH of the groundwater is near neutral to alkaline conditions (6.5 to 9). In this range, it is reported that the uranium and arsenic species are less reactive. So, what is the significance of this study if the species are already less reactive in natural conditions?
Author Response

(The authors gave the same response as above.)

Round 2
Reviewer 1 Report
I'd like to thank the authors for taking the time to respond to my original comments and the manuscript, in particular the clarity surrounding methods and the figures, has been significantly improved. However, I still find the lack of basic solid state characterisation concerning and disagree that these fundamental analyses (eg. XRD and/or SEM/EDX) are outside of the scope of the study.
Author Response
Dear Editor,
We sincerely thank you for your time and effort to process our manuscript. We are also grateful to the reviewer who has provided us with detailed comments, which allowed us to improve the quality of our manuscript. After careful examination of the reviewer’s comments, we have made revisions to the manuscript and prepared this point-to-point response. For reviewer’s inspection, the reviewer’s comments were repeated in below in Italic, whereas our responses were given in blue.
Reviewer’s Comment
I'd like to thank the authors for taking the time to respond to my original comments and the manuscript, in particular the clarity surrounding methods and the figures, has been significantly improved. However, I still find the lack of basic solid state characterisation concerning and disagree that these fundamental analyses (eg. XRD and/or SEM/EDX) are outside of the scope of the study.
Response:
We sincerely thank the reviewer for the constructive comments, which helped us improved the quality of this manuscript. Secondly, we did not include solid-phase characterization, for the following two reasons:
(1) the focus of this study is to investigate the reactivity between U(VI) and As(V), i.e. their tendency to form insoluble precipitates. With the comments from the reviewer, we have used 2 independent methods, which are wet-chemistry experiments and the model simulations, to prove that these two common ions are more likely to form insoluble precipitates at lower pH, not at higher pH. Therefore, we believe that the major points of this study have been proven.
(2) Characterization of the solids formed by U(VI) and As(V) is the topic of one of our on-going study. The characterization would include significantly more data. If we pack these data into this manuscript, then this manuscript would be lengthy for publication. We prefer to collect the solid characterization data and publish the whole body of data in our next manuscript as a whole.

Reviewer 2 Report
By and large, the authors have addressed the reviewers' comments satisfactorily. The manuscript may be accepted for publication after incorporating the following minor correction.
Comment 1: There are a few more typos and grammatical errors in the manuscript. Some of the sentences are not clear. For example, "the addition of nitric acid is to protect the sample before analysis and meet the requirements of ICP-MS." What is the protection the authors are referring to?
Author Response
Dear Editor,
We sincerely thank you for your time and effort to process our manuscript. We are also grateful to the reviewer who has provided us with detailed comments, which allowed us to improve the quality of our manuscript. After careful examination of the reviewer’s comments, we have made revisions to the manuscript and prepared this point-to-point response. For reviewer’s inspection, the reviewer’s comments were repeated in below in Italic, whereas our responses were given in blue.
Reviewer’s Comment
By and large, the authors have addressed the reviewers' comments satisfactorily. The manuscript may be accepted for publication after incorporating the following minor correction.
Comment 1: There are a few more typos and grammatical errors in the manuscript. Some of the sentences are not clear. For example, "the addition of nitric acid is to protect the sample before analysis and meet the requirements of ICP-MS." What is the protection the authors are referring to?
Response:
We sincerely thank the reviewer for the constructive comments, which helped us improved the quality of this manuscript.
(1) We have modified the section 2.3 according to the reviewer’s comment. We find the sentence that the reviewer mentioned redundant and removed this particular sentence. The section 2.3, after modification, is presented in below for reviewer’s inspection:
2.3. Measuring the concentration of U and As
The solution is exposed to air only during initial loading, pH adjusting and sampling. Samples were collected from the reaction vessel at pre-determined sampling time points. Samples for measurement of dissolved U, As were filtered using 0.22 μm polycarbonate membrane syringe filters. Then 50 μL of filtrate was taken, with the addition 250 μL of concentrated HNO3 for acidification, and 4.7 ml of ultrapure water for dilution. The concentrations of U and As in the diluted samples were measured using ICP-MS (Nexion 300, Pekin-Elmer, M.A., U.S.).
(2) We have proof-read the whole manuscript and polished the language for clarity.
